

# Solar SW radiative transfer in bubbled ice: spectral considerations, subsurface enhancement, and inclusions

Andrew R.D. Smedley[1,2], Geoffrey W. Evatt[2], Amy Mallinson[2], and Eleanor Harvey[2]

[1] School of Earth and Environmental Sciences, University of Manchester, Manchester, M13 9PL, UK
5 [2] School of Mathematics, University of Manchester, Manchester, M13 9PL, UK

*Correspondence to*: Andrew R.D. Smedley (andrew.smedley@manchester.ac.uk)

**Abstract.** We describe and validate a Monte Carlo model to track photons over the full range of solar wavelengths as they travel into optically thick bubbled ice. The model considers surface effects, scattering by bubbles and spectral absorption due to the ice. Using representative Antarctic ice bubble radii and number concentrations we calculate spectral albedos and 10 spectrally-integrated downwelling and upwelling radiative fluxes as a function of depth and find there is a marked subsurface enhancement in both the downwelling and upwelling fluxes relative to the incidence irradiance. This is due to the interaction between the refractive air-ice interface and the highly scattering interior and is particularly notable at blue and UV wavelengths which correspond to the minimum of the absorption spectrum of ice. A subsurface peak is also observed in the available radiative flux at depths of ~1 cm, and consequently the attenuation is more complex than can be described by a simple Lambert-15 Beer style exponential decay law. We find a moderate dependence on the solar zenith angle and surface conditions such as altitude and cloud optical depth. For macroscopic absorbing inclusions we observe geometry- and size-dependent self-shadowing that reduces the fractional irradiance incident on the inclusion's surface. Despite this the inclusions are subject to fluxes that are several times the magnitude of the single scattering contribution and act as local photon sinks. Such enhancement may have consequences for the energy budget in regions of the cryosphere where particulates are present near the surface. 20 These results also have particular relevance to measurements of the internal radiation field: account must be taken of both self-shadowing and the optical effect of introducing the detector.

## 1 Introduction

Incident solar radiation varies over a range of timescales due to the predictable seasonal and daily motion of the sun-earth system, supra-daily stochastic influences of the changing atmosphere and longer-term climate effects. It is a key driver of the 25 cryosphere's energy budget (van den Broeke et al., 2011; Van Tricht et al., 2015; Hofer et al., 2017) and its variability effects internal temperature profiles of ice, particularly close to the surface (Liston et al., 1999). This has strong implications for processes such as ice sheet near surface melting and ice shelf crack formation (Bennartz et al., 2013; van den Broeke et al., 2016; Webster et al., 2017). In addition to driving physical processes, the quantity and spectral composition of solar radiation transmitted through, or available within, ice can have significant effects on aquatic and other polar ecosystems.





Specifically, the present study was initiated in response to the need for a better understanding of this glacial subsurface radiative field. Recently Evatt et al. (2016) presented a mathematical model for the movement of meteorites through blue ice in Antarctica. In this work the attenuation of solar radiation through ice and the absorption of solar radiation by a meteorite were modelled using the Lambert-Beer law. Although this approach works well at certain depths, it has limitations, becoming

inaccurate particularly near the surface. This issue specifically motivated us to find a more accurate, yet still simple and easily applicable, methodology for modelling the attenuation and absorption of solar radiation through blue ice.

Despite this specific motivation, there are obvious parallels to be drawn with several different climatologically important research foci. Examples include the impact of anthropogenic soot, pollutants, cryoconites and other englacial absorbers near the surface of the Greenland Ice Sheet (e.g. Dumont et al., 2014; Box et al., 2012; Stibal et al., 2017). In all of these cases,

including that of meteorites, the inclusions have a low albedo and are subject to an atmospherically-modulated near-surface radiation field, cause local heating and thus can contribute to increased melt rates.

Notwithstanding the importance of shortwave (SW) radiative transfer in the aforementioned studies there is a tendency, as in Evatt et al. (2016), to treat the shortwave radiative flux as a single broadband parameter (via the Lambert-Beer law) and neglect to incorporate the range of behaviours exhibited by different wavelengths of solar radiation. This is a fundamental

simplification as the incident solar spectrum exhibits a great deal of structure due to terrestrial and solar processes, whilst the absorption spectrum of ice spans eight orders of magnitude across the solar wavelength range (Warren and Brandt, 2008). For some applications such a simplification is reasonable, however for others – such as those concerned with the near-surface heat budget – a more encompassing model will be beneficial.

There are three key studies that have investigated radiative transfer within ice in detail and gave due attention to these spectral

issues. Mullen and Warren (1988) developed a radiative transfer model of lake ice to illustrate the processes responsible for the resulting albedo and transmission through a layer of ice. They treated bubbles as spheres, deriving the scattering coefficient and asymmetry parameter from Mie calculations, and relied on the delta-Eddington method in their treatment of multiple scattering. They showed results across the solar waveband for the direct beam and diffuse incidence, and the transmission for different bubble concentrations. Later, Light et al. (2003) developed a Monte Carlo model for radiative transfer in sea ice.

Their focus was on cylindrical samples of ice, with the model being used to interpret backscattering from cylindrical core samples. In both these studies the emphasis is on relatively optically thin samples where the ice overlays a body of water, or where the parameter of interest is the transmission. In cases where ice overlays water there is very little change in the refractive index at the lower ice-water interface: this effectively removes any lower refractive boundary and permits downwelling photons to continue their trajectory into the water with reduced opportunities for further scattering. Therefore an assessment

of the transmission and reflectance of the incident sunlight is considered an adequate summary of the interaction in these cases. The third key study presented by Liston et al. (1999) relates most closely to that described here, detailing the ice melt in blue ice vs. deep snow areas. Their focus, however, is on understanding the resulting subsurface temperature and melting profiles, and they rely on a two-stream approach which is constrained by specific atmospheric forcings and measured surface albedos. Account is taken of the spectral nature of the problem; however to maintain consistency between the treatment of snow and



blue ice the optical properties are linked to effective grain size and a spectral extinction coefficient is calculated on this basis. We also note related studies that parameterised the spectral albedo of white sea ice and snow, but not the internal radiative field (Malinka et al., 2016; Gardner and Sharp, 2010), or considered internal scattering from a purely theoretical perspective (Malinka, 2014).

However to our knowledge there has been no study for optically thick blue ice where the spectral radiative transfer and albedos are derived from the interaction with the embedded bubbles and the underlying material properties. In the present study we therefore take an alternative approach in order to fill this gap and thus address two core aims. The first is to present an in-depth investigation of the radiation field within optically thick bubbled ice at different solar wavelengths, including a range of sensitivity tests and its impact on inclusions. This then leads us to the second aim: a distillation of these results into a simple,
and widely applicable, mathematical model for the net flux.

In Sect. 2 we describe the details of our Monte Carlo model, including its initialisation, the conditions at the boundaries, the scattering of photons by bubbles and the eventual absorption of the bubbles into the ice. The validation of the model is also discussed. In Appendices A and B we describe our methodology for the related calculations of the incident solar spectrum and the derivation of the relevant bubble parameters. In Sect. 3 the model is used to investigate how the incident solar spectrum
propagates through the ice, both when considered spectrally and when integrated across solar wavelengths. We investigate the influence of varying the bubble number concentration and effective radius, the effect of varying the solar zenith angle and the influence of different surface environments which might alter the spectral balance of the incident solar spectrum. In Sect. 4 a macroscopic inclusion (absorbing target) is added to the model in order to study how the target's geometry and size impact on the effective radiation field incident on its surface. Lastly, in Sect. 5, curves are fit to the integral shortwave results for the net
radiative flux in a typical blue ice area.

## 2 Monte Carlo model description

Whilst different approaches to utilising the general radiative transfer equation (for background see Thomas and Stamnes, 1999) exist, here we choose to use a Monte Carlo simulation approach to investigate radiative transfer within bubbled ice. We do so for its ability to represent the different physical aspects and its intuitive nature.

The core Monte Carlo model aims to calculate the downwelling and upwelling solar irradiance fluxes, $E_\downarrow$ and $E_\uparrow$, as a function of depth through bubbled ice by tracking the random walk pathway of simulated photons through the medium. Photons incident on the ice surface arrive from the atmosphere, and if not reflected by the interface, pass into the ice. Photons entering the ice interact both with the ice through absorption and with the bubbles trapped within it by scattering. Macroscopically the ice is considered as a semi-infinite slab with no lower boundary. Within this slab we follow the approach of Mullen and Warren
(1988) and model a distribution of bubbles of effective radius $r_{bub}$ and number concentration $N_{bub}$. Figure 1 shows a schematic of the model geometry. Our initial focus is on the downwelling and upwelling irradiance as they encapsulate more information about the internal radiation field than does the net irradiance, or the absorption profile with depth, whilst still allowing these





quantities to be derived as necessary. However the net irradiance (flux) is clearly important in a glaciological framework and we will turn our attention to it in Sect. 5. Throughout we define the fractional irradiance as the fraction of photons arriving at the surface that are in flight and contribute at a given depth below the surface, whether upwelling or downwelling: $E_\downarrow/E_0$ and $E_\uparrow/E_0$, that is, normalised by the incident surface irradiance, $E_0$. Accordingly absolute values of the upwelling and

downwelling components can be calculated by multiplying the fractional quantities by the incident surface irradiance. Likewise the spectral albedo is defined at each wavelength, $\lambda$, as the fraction of incident photons that escape back to the atmosphere, either by direct reflection or internal scattering; this is calculated under a diffuse sky except where noted otherwise.

## 2.1 Model inputs

There are three principal sets of inputs to the Monte Carlo model: the incident solar spectrum, the intrinsic optical properties

of bubbled ice, and the geometric characteristics of the bubbles. These are detailed in turn below.

The incident spectral irradiance at the ice surface is calculated using the libRadtran radiative transfer model (Mayer and Kylling, 2005) with relevant atmospheric inputs, for clouds, aerosols, and solar zenith angle. These, the surface altitude and broadband albedo are initially chosen to be appropriate for a blue ice area near the Frontier Mountain range, Antarctica [72.95° S, 160.48° W]. The inputs are similar to those Evatt et al. (2016) used to calculate a climatology of integrated SW fluxes,

however for completeness they are described in some more detail in Appendix A.

The intrinsic optical properties of the ice are fully defined by the wavelength of the photon being tracked ($\lambda$), $r_{bub}$, and $N_{bub}$ as follows. The complex refractive index is taken from Warren and Brandt (2008), with the real part, $m_{re}$, being interpolated at intermediate wavelengths ($\lambda$) linearly in log($\lambda$), and the imaginary part, $m_{im}$, being interpolated to intermediate wavelengths on a log-log basis. The scattering and absorption coefficients, $k_{sca}$ and $k_{abs}$, are then expressed in terms of the complex

refractive index, the bubble effective radius, and the number concentration, as follows (Muller and Warren, 1988; Hecht, 2002):

$$k_{sca} = \frac{2\pi r_{bub}^2}{N_{bub}}, \tag{1}$$

$$k_{abs} = \frac{4\pi m_{im}}{\lambda}. \tag{2}$$

Turning to the geometric characteristics of blue ice bubbles, there is a paucity of relevant in-situ data and thus we principally

rely on a homogenisation of bubble density and radii concentration measurements carried out by Dadic et al. (2011; 2013) near the Transantarctic Mountains, Antarctica. In summary Dadic et al.'s collected samples cover a range of ice conditions along two BIA transects; cores from depths down to 0.94 m were analysed by a combination of microCT analysis, specific surface area (SSA) derived estimates and caliper-and-scale measurements. We have carried out a homogenisation exercise of these and observations from the wider literature to determine sets of internally consistent parameters linked to bulk properties,

including the effective scattering coefficient to retain generality. Those referred to further are shown in Table 1; more details are provided in Appendix B.



## 2.2 Photon Initialisation

Individual photons are tracked from their incidence on the air-ice interface ($z = 0$) until they return to the atmosphere, are absorbed, or pass below a depth where their contribution is negligible, here, $z = -16$ m. Each photon is initialised with a Cartesian position and unit direction vector. The $x$ and $y$ coordinates are assigned as random numbers in the range [0, 1]

corresponding to a 1 m × 1 m square on the upper interface, with the initial $z$ coordinate being set to zero. To simulate photons from a diffuse sky (or its diffuse component) the direction vector, expressed as direction cosines, $[\mu_{x0}, \mu_{y0}, \mu_{z0}]$ is initialised as a random point on a unit sphere chosen by generating three Gaussian random variables, $[x, y, z]$, which are then normalised (Muller 1959, Marsaglia 1972):

$$\begin{bmatrix} \mu_{x0} \\ \mu_{y0} \\ \mu_{z0} \end{bmatrix} = \frac{1}{\sqrt{x^2+y^2+z^2}} \begin{bmatrix} x \\ y \\ z \end{bmatrix}. \tag{3}$$

Those photons initialised with a positive $z$ coordinate direction are then negated to ensure they represent an initial downward direction. For the direct solar beam component, the direction vector is set appropriately.

Specular reflections at the surface are dealt with by calculating the wavelength and angular dependent reflection coefficient for an unpolarised beam according to Fresnel's equations of reflectance (Hecht, 2002). If a random number in the interval [0, 1] is less than the calculated reflection coefficient, the photon is marked as being returned to the atmosphere. If it is greater,

the photon is considered to have passed into the ice and the direction vector is updated in line with the angle of refraction.

## 2.3 Photon Interactions

For photons that have passed into the ice the next stage is to calculate the distance before a scattering or absorption event occurs. Accordingly we construct two cumulative exponential distributions with mean free paths $k_{abs}^{-1}$ and $k_{sca}^{-1}$, and sample randomly from these in the interval [0, 1]. Whichever produces the smallest result, equivalent to the shortest distance travelled,

is the event that is considered to have occurred. In either case, the photon position is updated from the direction of travel and distance.

The updated position is checked at each iteration against the local boundaries. First the $x$ and $y$ coordinates are constrained to stay within the limits set at the surface by applying a periodic boundary condition: $x_{i+1} = x_i \bmod 1$, and likewise for $y$. Further if the updated photon position exceeds a depth of 16 m, it is flagged as such and no longer tracked. If the updated position is

found to lie above the ice-air interface, a random number on the interval [0, 1] is compared to the Fresnel reflection coefficient to determine whether it will be transmitted or reflected. If transmission to the atmosphere occurs, the photon packet is marked as such and no longer tracked. If it is reflected back down by the surface, the $z$ coordinate of its position and direction are negated and tracking continues.

If the event corresponds to absorption this is flagged and the photon is no longer tracked. If the interaction is a scattering event

then a new direction of travel is calculated, defined by a deflection angle and an azimuthal angle w.r.t. the original direction of travel. The deflection angle is calculated using a Henyey-Greenstein phase function (Henyey and Greenstein 1941) with the



dependent variable, the asymmetry parameter $g$, calculated from Mie theory (Bohren and Huffman 1983) (Fig. 2). Following Pulli et al. (2013) the deflection angle, $\theta$, is calculated from

$$\cos \theta = \frac{1}{2g}\left[1 + g^2 - \left(\frac{1-g^2}{1-g+2g\chi}\right)^2\right], \tag{4}$$

where $\chi$ is a uniformly distributed random number in the interval [0, 1). The rotationally symmetric azimuthal angle, $\phi$, is

calculated as

$$\phi = 2\pi\chi. \tag{5}$$

From these two angles (which define the scattering event) and the original direction vector, an updated direction cosine vector is calculated. The process described in this sub-section is repeated until each photon is returned to the atmosphere, passes below the lower boundary, or is absorbed. Whilst for conceptual reasons the algorithm has been described above as following

a single photon, in the model we make use of MATLAB's vectorisation capabilities and track typically $10^4$ photons at a single wavelength, using array indexing only to advance the positions of those that are still in flight.

**2.4 Photon Counting**

As well as recording the final positions of the photons we track them throughout their multiply scattered passage through the ice. In order to do so we record the depth of each photon and whether the direction of its flight is positive (downwards) or

negative (upwards) at every step. This allows us to determine the downward and upward irradiance fluxes respectively as a fraction of the total photons that were initially released. In addition to this the $z$ coordinates of absorbed photons can be used to calculate the fractional absorption as a function of depth (here only used as an internal consistency check) and the total fraction of photons returned to the atmosphere (the albedo). We repeat the process at wavelength intervals of 10nm from 280nm to 2800nm. For results integrated over the complete solar shortwave band, the single-wavelength results are interpolated

to a 1nm grid, and then weighted by the incident spectral irradiance, before summing. Note also that whilst the model internally uses a $z$ coordinate that becomes more negative with distance into the ice, we treat the depth, $d$, as positive parameter from section 3 onwards, increasing below the surface.

**2.5 Model Validation and Limitations**

To validate the described model and test its predictive skill we follow the example of Light et al. (2003) who relied on the

four-stream results of Grenfell (1991). For the outputs of interest to us there are three key scenarios in common. The first is a conservative non-refractive domain where we calculate the albedo and transmissivity of a horizontally infinite slab at a range of optical depths, $\tau$, and asymmetry parameters, $g$. Next, a conservative refractive case where we add a lower boundary to our model and set the refractive index of the slab to be 1.31. Finally, a refractive non-conservative case, where we use Light et al. (2003)'s values of $k_{abs}$ as noted in their table 2, and, as in the former two cases, choose all other inputs to match those used

by Light et al.




Doing so we find the discrepancy between the albedo and transmissivities calculated with the method described herein and the four-stream solution in Light et al. are typically ~0.5 % in all three cases. The albedo results for the conservative non-refractive and refractive cases are shown in Fig. 3a and Fig. 3b respectively.

Internally we also check the reproducibility of repeated runs to ensure a stable solution has been reached. For an example wavelength of 600 nm the choice of tracking $10^4$ photons produces $E_\downarrow$ profiles that are consistent at the level of 0.75% (the standard deviation of 5 repeated runs, averaged over depths from 0 m to 2 m). When calculating broadband parameters a total of $2.53 \times 10^6$ photons are tracked, and accordingly this value is found to reduce to 0.041% over the same range of depths.

Whilst the Monte Carlo model produces reproducible results, shows good agreement across the range of optical parameters used by Light et al., covering the range of absorption and asymmetry parameters exhibited by blue ice areas, there are some limitations in regards real-world applications. First of these is that all bubbles are assumed to be spherical and thus their single scattering behaviour is governed by Mie theory. This is a useful simplification but in reality individual samples and individual bubbles within them will deviate from perfect spheres. This will affect the asymmetry parameter, and consequently the observed attenuation, but to answer the broad question we choose not to further specify the bubble geometry beyond the assumption that it is spherical. Neither do we consider any vertical or spatial inhomogeneity of bubble density that clearly exist, but have chosen to determine typical values that are relatable to bulk ice parameters in order to maintain the general applicability of the results. We note that vertical variations in the asymmetry parameter, the effective bubble radius, and their density could be dealt with by relatively small adaptations to the code.

The model also assumes that the ice-surface is planar, and does not account for any partial covering by a wind-blown snow layer. On the whole blue ice layers are largely free of snow, but we estimate that an intermittent snow layer of depth 5 cm, covering approximately 10 % of the surface would reduce the visible irradiance incident on the upper ice surface by 5 % to 10 % on average (Perovich, 2007). Counteracting this we also anticipate a similar degree of enhancement of the incident irradiance under cloudy skies from albedo feedback that may have not been captured by the use of a broadband albedo input to the atmospheric radiative transfer calculation and by not including any partial snow layer in the model.

## 3 Monte Carlo model results

Our modelling assumes a prescribed description of the incoming solar irradiance spectrum, and estimates of the bubble number concentration and effective radius. In this study we use a solar irradiance based upon the Frontier Mountain Blue Ice Area, Antarctica [72.95° S, 160.48° W]; full details of its calculation are provided in the Appendix A. Suitable independent estimates for Antarctic Blue Ice Area bubble concentrations and effective radii are stated in Table 1, and background information regarding their calculation is provided in Appendix B.



### 3.1 Spectral considerations vs. depth and irradiance enhancement

Assuming fully diffuse sky conditions (that is the incident irradiance has no separate direct beam component), we now apply our Monte Carlo model to four air bubble parameter sets as detailed in Table 1.

In all cases the spectral attenuation by ice is controlled by the interaction of the spectrally-independent scattering coefficient,

$k_{sca}$, and the absorption coefficient in pure ice, $k_{abs}$. As seen in Fig. 4, the latter is strongly wavelength dependent, varying from between $10^3$ m$^{-1}$ and $10^5$ m$^{-1}$ for $\lambda > 1430$ nm, to a minimum of $6.4 \times 10^{-4}$ m$^{-1}$ at $\lambda = 390$ nm. Consequently we observe the well-known rapid absorption of photons at the longest wavelengths first, and a shift of the peak in the attenuated spectrum towards shorter wavelengths (Fig. 4). For the no cracks parameter set, corresponding to the least scattering ($k_{sca} = 102.2$ m$^{-1}$) and the largest mean free path of 0.01 m, the solar signal is reduced below 1 mW m$^{-2}$ nm$^{-1}$ for all solar $\lambda > 1351$ nm by a

vertical depth of 0.01 m.

Notably for UV and the shortest visible wavelengths there is an enhancement of the subsurface downwelling irradiance, $E_\downarrow$. Intuitively this is unexpected as the downwelling irradiance is greater than that incident on the ice surface, but it has been noted in literature previously (Jiang et al., 2005). The enhancement is a result of the change in refractive index at the air-ice interface, combined with the higher refractive index medium exhibiting both scattering and relatively low absorption. Photons

that enter the ice are scattered by air bubbles (or other contaminants), and assuming a low absorption coefficient, eventually return to strike the air-ice interface. At this point photons arrive at the interface from the high refractive index medium, and those with an incidence angle greater than the critical angle will undergo total internal reflection, contributing a further time to the downwelling irradiance. Providing it is not absorbed, a photon will continue to be scattered within the ice and may be reflected repeatedly by the inner surface, contributing multiply to the downwelling irradiance. Likewise the upwelling

irradiance will be correspondingly enhanced, and thus the net solar flux across the ice-air interface does not change and energy is conserved (as noted in Jiang et al., 2005). The energy available for absorption by small contaminants, inclusions, or heating of the ice itself are however increased by this enhancement process.

To provide an upper limit for the enhancement we assume a semi-infinite refractive ice slab that scatters but does not absorb photons. For photons incident on the upper surface of the ice-air interface a fraction, $T$, will be transmitted into the body of

the ice and contribute to the downwelling irradiance. In the absence of absorption all these will return to impact on the inner side of the interface. A fraction $R$ of these will then reflected downward to contribute a second time. Extending this argument to multiple reflections we find a maximum enhancement factor of:

$$\sum_{n=0}^{\infty} TR^n = \frac{T}{(1-R)} . \tag{6}$$

To evaluate this expression, we assume diffuse incident radiation fields on both the upper and lower surfaces of the interface

and apply Fresnel reflection and transmission coefficients; for a real refractive index of ice of 1.306, this gives $T = 0.832$ and $R = 0.668$. Consequently the downwelling irradiance can be increased by a factor of up to 2.506 times the incident irradiance at the ice-air interface. For all UV and visible wavelengths the potential enhancement is $\geq 2.499$. For longer



wavelengths the absorption by pure ice increases and multiple scattering, and with it the enhancement process, is greatly reduced.

Our Monte Carlo results fall within these theoretical limits (Fig. 4), with a maximum enhancement factor of 2.13 at $\lambda = 410$ nm and a depth of 0.06 m. At 550 nm the enhancement factor is reduced to 1.743 due to the increasing absorption. In comparison Jiang et al. (2005) noted an enhancement factor of 1.32 at 550 nm, but this was for a slab 1.7 m thick underlain by sea water and with a lower mean porosity of $< 1$ %. Both losses through the lower boundary and reduced single scattering albedo ($k_{sca}/(k_{abs} + k_{sca})$) reduce the enhancement.

**3.2 Effect of bubble parameters**

The same interaction between the scattering coefficient and the absorption coefficient that controls the wavelength-dependence of irradiance also results in a strong wavelength dependence in the total albedo. There are two contributions to the overall albedo of blue ice: the direct, specular reflection according to Fresnel and the contribution from internally back-scattered photons that return to and escape from the surface. The first only exhibits a weak dependence on wavelength, but the latter relies on photons entering the ice not being absorbed before they travel sufficiently far to be scattered back to the surface. Consequently for $\lambda > 1440$ nm the albedo has no internally scattered component, but for shorter $\lambda < 400$ nm the low absorption coefficient results in an albedo that rises to within ~2 % of unity (Fig. 5). Though scattering by bubbles is a necessary part of the process, we note that a factor of four increase in $k_{sca}$ alters the albedo at 820 nm from 0.36 to 0.52, an increase of less than 50 %. As a result the solar irradiance weighted (or broadband) albedo is also only weakly dependent on the scattering coefficient, varying from 54.6 % for the no cracks parameter set, to 63.7 % for the lower parameter set based on Bintanja's (1999) 850 kg m$^{-3}$ density end point; the solar-irradiance weighted albedo for the mean dataset is 62.3 %. The range of these results (55 % to 64 %) includes the albedo estimate used as an input to the libRadtran calculations of 62 %, and notably the albedo for our mean dataset agrees well with this initial estimate. The range of values also compares favourably with field measurements: Bintanja (1999) quotes an observed range of 56 % to 69 %, whilst Dadic et al. (2013) cites an overall range of 55 % to 65 % from several earlier BIA studies. In contrast Dadic et al's own observations from three specific BIA locations (mean of clear sky and cloudy albedos) show a shift to higher values (61 % to 67 %).

From a visual perspective it is notable that the spectral variation in the albedo implies that the characteristic colour of bubbled ice is predicted (Fig. 5): high albedos are found at wavelengths associated with the human eye's short-wavelength (blue) cone response, reducing at visible wavelengths associated with the green cone response, and then still further at wavelengths where the long-wavelength cone is preferentially sensitive (red). In addition, this reinforces the point that blue ice environments with smaller scattering coefficients, $k_{sca}$, (due to fewer or smaller bubbles) result in the most saturated colours, and consequently, show increased fluxes below the surface.

The wavelength-integrated results are shown in Fig. 6, normalised by the downwelling irradiance incident on the surface, $E_0$, which here has been calculated to be 302.5 W m$^{-2}$. Though we observe the expected quasi-exponential fall off with depth, with greater attenuation for cases with increased scattering, the Monte Carlo results also predict the presence of a sub-surface peak



in the upwelling flux at depths of ~1 cm, a feature that is still observable in the fractional mean irradiance, $(E_\downarrow + E_\uparrow)/2E_0$, as shown in Fig. 6d. This sub-surface peak is interpreted as being a result of the downwelling enhancement below the surface: immediately above the surface region there is a comparatively reduced downwelling irradiance and thus a reduced contribution via backscattering to the upwelling irradiance. Only at slightly lower depths $\sim 1/k_{sca}$ does the backscatter contribution

correspond to the peak in the downwelling flux. We also note that the upwelling flux is a substantial fraction of the downwelling flux, almost reaching parity below depths of 1 m. In general bubble parameter sets with reduced scattering (smaller $k_{sca}$) produce a less-defined sub-surface peak and increased solar flux at depth, though the dependence on scattering coefficient is weak.

### 3.3 Dependence on SZA and geographic location

The results presented so far have relied on the assumption that the incoming solar flux arrives from a diffuse hemispherical sky (with no direct component) and assumed atmospheric and other inputs selected according to a specific location, the Frontier Mountain Range, Antarctica. To test the sensitivity of the presented results to differing assumptions we have rerun the Monte Carlo model to include partitioning between direct and diffuse solar components and at a range of solar zenith angles, and additionally, with surface elevations, cloud optical depths and solar zenith angles appropriate for a range of geographic

locations across both polar regions. In short we find that once partitioning between diffuse and direct components is accounted for, the attenuation of the downwelling and upwelling fluxes show a weak dependence on SZA, though this is more obviously expressed in the spectral albedo at IR wavelengths. Likewise there is only a moderate dependence when specific geographic inputs are chosen – the most obvious effect is a reduction in the IR part of the incident solar flux for locations which exhibit large cloud optical depths or airmass factors. Accordingly, higher latitude, lower elevation and cloudier locations experience

somewhat reduced attenuation of the downwelling and upwelling fluxes per unit of incident flux. This a secondary effect outweighed by the presence of a cloud layer that will reduce the total irradiance incident on the ice surface.

Further details can be found in the supplementary material.

### 4 Inclusions

So far the Monte Carlo model has been used to investigate the impact of varying bubble radii, number concentrations and the

solar zenith angle on the propagation of solar irradiance into blue ice; we now apply it to investigate the energy that impacts upon and is absorbed by inclusions within the ice. Accordingly we adjust the model to count photons whose path intersects with a defined volume element that represents the inclusion. For computational reasons we restrict this test to photons whose position falls within a set distance of the centre of the inclusion. The path between one scattering event and the next is then subdivided at a granularity of <1mm depending on target size and the photon is treated as absorbed if any point along this path

lies within the inclusion volume. The absorbed photon is added to the downwelling or upwelling count as appropriate.



To model a range of possible inclusions we model spherical, planar and ellipsoidal geometries. Following Evatt et al. (2016), we choose dimensions appropriate for englacial meteorites augmented by one smaller and one larger geometry (see Appendix C for further details). A single inclusion is defined during each model run to ensure independence: we calculate the fractional downwelling and upwelling irradiance incident on the inclusion at 10 geometrically spaced depths.

Figure 7a shows the fractional irradiance incident on the set of spherical inclusions whilst Fig. 7b shows the same for planar inclusions. The most prominent feature is that for both cases the fractional irradiance per unit surface area is substantially lower than without an inclusion. We attribute this to self-shadowing of the diffuse radiation field. That is, downwelling photons once absorbed by the inclusion cannot be scattered up, and then down to contribute multiply to the irradiance. Despite this the fractional irradiance absorbed by the inclusions is still markedly greater than the singly scattered contribution (also shown on

Fig. 7a and 7b), the inclusion acting as a sink for photons in its vicinity. This self-shadowing can also be seen to be dependent on both the geometry and dimension of the inclusion: when a target has larger extent in the x-y plane compared with the scattering length, it becomes increasingly unlikely for photons from one side of the inclusion to impact the opposing surface, and, consequently, the mean irradiance incident on the target is reduced. For a large inclusion close to the surface the downwelling irradiance is similar to the transmitted (singly scattered) fraction of the incident solar signal as few additional

upwelling photons can enter the region between the air-ice interface and the upper surface of the inclusion. For smaller inclusions near the surface the downwelling irradiance increases above the singly scattered contribution, but to a lesser degree than at depth.

For absorbing inclusions, the energy balance may lead to the surrounding ice reaching melting point. Once it does any inclusion denser than water will move downwards under gravity and be capped by a water layer above. The air previously trapped in

bubbles for porosities > 2.9 % (Battino et al., 1984) will form a layer above the meltwater and create a pair of interfaces: one ice-air and one air-meltwater. These two interfaces reflect a fraction of the diffuse downwelling irradiance that would have otherwise reached the inclusion. Taking a solar-spectrum weighted refractive index of $m_{re} = 1.3061$ and applying Fresnel reflection coefficients, we find that 67.8 % of the diffuse downwelling irradiance field will be transmitted from the ice and into the air. At the air-water interface 83.1 % of this is further transmitted into the meltwater where it will continue unimpeded

to the inclusion. In total the presence of an air layer could reduce the downwelling irradiance incident on the inclusion to 56.3 % of the expected value, though its' horizontal extent would also have to be taken into account.

The processes of total internal reflection by an ice-air boundary within the ice and self-shadowing also relate to the measurability of the predicted irradiance fluxes. Specifically attempts to measure the irradiance by insertion of an optical detector into the ice would have to account for both these points. The degree of self-shadowing would be a function of both

the size and geometrical shape of the detector, whilst the transmission across the interfaces between the ice, a (partial) air layer, and the outer envelope of the detector would need to assessed carefully. If they were not included, the irradiance fluxes within the body of the ice would be under-estimated.



## 5 Comparison to analytic solutions and curve fits

Whilst the described Monte Carlo model aims to replicate the radiative transfer processes occurring within the ice accurately, it is computationally intensive. For each combination of bubble parameter sets and SZA, we follow and track the paths of approximately $2.53 \times 10^6$ photons, which requires ~15 hours of runtime on a modern desktop PC. Accordingly there is a utility to being able to replicate these results, but by analytical means.

Following Marchesini et al. (1989) we can define an effective attenuation coefficient, $K_{eff}$, for a medium where both anisotropic single scattering and absorption are in operation:

$$K_{eff} = \sqrt{\left(3k_{abs}^2 + 3k_{abs}k_{sca}(1-g)\right)}, \tag{7}$$

where the other symbols have the same meanings as previously. In the low scattering limit, this reduces to:

$$K_{eff} = \sqrt{3}k_{abs}, \tag{8}$$

whilst the $\sqrt{3}$ multiplier can be understood as the inverse of the direction cosine, $\bar{\mu}_z$, which has been averaged over each hemisphere as appropriate for an isotropic radiation field. In Fig. 8 we plot the spectral variation of $K_{eff}$ alongside the e-folding distances derived from the Monte Carlo results for the Dadic BIAs no cracks parameter set and a diffuse solar irradiance. However the near surface downwelling irradiance field is not isotropic as assumed by Eq. (8), but primarily lies within a vertically-orientated cone defined by surface refraction, which implies a larger mean value of $\bar{\mu}_z$, In line with this we note better agreement between the values of $K_{eff}$ from Monte Carlo e-folding distances and calculated via Eqn. 8 at $\lambda >$ 1200 nm, where absorption dominates scattering, if the first multiplier is reduced to ~2 to account for this more directional field (Fig. 8). For single wavelengths this construct gives good agreement with the Monte Carlo results, save for the shortest wavelengths and the largest absorption coefficients where discrepancies occur due to the vertical grid interval and overall size of the model domain.

In light of this, it is tempting to consider the form of the spectrally integrated attenuation curves in Fig. 6c as quasi-exponential, and following the well-known exponential relation given by the Lambert-Beer law for attenuation through an absorbing medium. The Lambert-Beer exponential relation holds at a single wavelength: longer wavelengths exhibiting a high attenuation, and shorter wavelengths having a lower attenuation. Using an integrated form of the Beer Lambert decay function is common practice (Cuffey and Paterson, 2010; Evatt et al., 2016), with attenuation values typically around 2.5 m$^{-1}$, as this leads to mathematically tractable estimates for ice temperatures and melt rates (Evatt et al., 2016). However this approach is somewhat crude and does not explicitly account for the absorption differences in wavelength, e.g. Fig. 8. To help overcome this, Bintanja (1999; 1997; 2000) used a model which assumed all of the shorter wavelength energy was absorbed at the surface, and only the longer wavelengths were able to penetrate to greater depths through an exponential relation. Yet this approach automatically fails to provide explicit information as to the irradiance in the uppermost few centimetres of the ice. One might be tempted to overcome this through use of a double exponential, with the two exponential coefficients being representative for short and long wavebands. However the absorption coefficients do not fall into two clear ranges and we





therefore expect a double exponential fit to underestimate the solar flux at intermediate depths, and conversely to over-estimate the flux at smaller and greater depths. As such, we suggest a triple exponential function to approximate the solar energy flux attenuation. We now describe how such a function can be parameterised.

As seen in Sect. 3, the attenuation of the irradiance depends upon bubble size and distribution. If the attenuation for distinct ice samples was highly different, then before an analytical approximation could be found, one would first have to solve the full presented Monte Carlo model. Fortunately, plots of the irradiance against depth for the results of Fig. 6c, when scaled against their own surface albedo, show very similar attenuation profiles (Fig. 9) — clearly the result holds less well for the no cracks data set. As such, one need not necessarily run the Monte Carlo code for each new ice sample. Instead, assuming the ice in question is reasonably generic, one can use a triple exponential function that has been best-fitted to the scaled data sets presented here. For example, if the Dadic BIAs no cracks data is omitted, then the mean scaled net downwelling irradiance, $M_f = \|E_\downarrow - E_\uparrow\|$, has the curve fit against depth, $d$:

$$M_f(d) = 0.479e^{-18.1d} + 0.113e^{-2.04d} + 0.409e^{-695d}. \tag{9}$$

Here all of the coefficients sum to unity so that, if the albedo, $a$, and incoming solar irradiance, $Q$, is known, the net irradiance, $N_f = E_\downarrow - E_\uparrow$, at a depth, $d$, can be approximated as:

$$N_f(d) = (1 - a) \cdot Q \cdot M_f(d). \tag{10}$$

In so doing, the unique information regarding bubble size and distribution is still present within this equation, for it manifests itself in the size of the albedo, $a$ (where surface albedo estimates are commonly collected field measurements). It is interesting to note that the slowest attenuation coefficient within $M_f$ is, at –2.04, close to the value of –2.5 commonly used by Bintanja et al. (1997; 1999; 2000) and somewhat smaller in magnitude than the –3.3 assumed by Liston et al. (1999) in their constant bulk extinction coefficient simplification.

## 6 Summary

In this study we have described the formulation of a Monte Carlo model which calculates the shortwave radiative transfer in optically thick ice where englacial bubbles cause scattering. Our results are primarily applicable to blue ice areas where the surface can be considered horizontal and the ice is relatively compact and snow-free. However the general conclusions are expected to be relevant for optically thick glacial ice where scattering dominates, though the internal radiative processes can be complicated by the microscopic geometries of snow and firn as well as the macroscopic surface geometry. Notwithstanding this simplification, the general results that follow are expected to have wider importance.

Our first main finding is that there can be an appreciable enhancement of the subsurface downwelling flux, above the level of the incident irradiance (previously observed by Jiang et al. (2005) for a single wavelength of 550 nm). For normalised units of irradiance that we have used throughout, the integrated solar downwelling flux can be as high as 1.4. There is a corresponding enhancement in the upwelling flux (thus conserving energy). This enhancement is a result of the refractive air-ice interface overlaying a volume where scattering dominates, resulting in multiple scattering and internal reflections. For wavelengths



where absorption is minimal the enhancement can be as high as 2.13, compared to a theoretical maximum of approximately 2.5 for ice at solar wavelengths in the absence of absorption. We also note subsurface peaks in the upwelling flux at depth ~1 cm due to the interaction of the downwelling enhancement and backscattering. These features, and the wide range of absorption coefficients exhibited at UV and blue wavelengths, in contrast to at IR wavelengths, result in a depth dependence that is

inadequately modelled as a single exponential.

Considering the spectral albedo, our calculations produce the expected wavelength dependence that is interpreted by our visual system as 'blue', with lower porosity ice producing more saturated colours. The albedo is generally insensitive to changes in the amount of scattering, even at 820 nm, where the effect is largest, we see only a 50 % change in the spectral albedo when $k_{sca}$ changes by a factor of four. In addition our model demonstrates the two components that contribute to the overall albedo:

the direct Fresnel surface contribution that shows a weak dependence on wavelength, and the strongly wavelength dependence from photons entering the ice and being scattered until they return to the atmosphere.

Our results are predicated on a diffuse incident field under particular assumptions of the surface environment. However, permitting the solar zenith angle to vary, we find only a moderate dependence once the incident field is partitioned between diffuse and direct incident components. Likewise, considering a range of polar surface environments, the predominant effect

is a reduction of the incident radiative field at IR wavelengths for locations which exhibit large cloud optical depths, or, for clear sky cases, airmass factors. As a result, higher latitude, lower elevation and cloudier locations show somewhat reduced attenuation of the normalised downwelling and upwelling fluxes.

For absorbing inclusions embedded within the ice and its internal radiation field, self-shadowing reduces the irradiance incident on the surface of the inclusion. This is a geometric effect, the irradiance reducing for larger inclusions. Conversely smaller

inclusions whose dimensions are less than the mean free path for scattering absorb a greater fraction of the available radiation; for all inclusion sizes downwelling and upwelling fluxes lie between the available irradiance and the single scattering component, the inclusion acting as a local sink for photons. Interpreting the inclusion instead as an optical detector we conclude that both self-shadowing and the introduction of lower refractive interface must be taken into account when assessing measurements.

Finally we assess the results presented in Sect.3 as a whole, and given the moderate dependence on the attenuation to the solar zenith angle, environmental conditions and bubble parameters, formulate an empirical expression describing the typical behaviour of the net downwelling irradiance in optically thick blue ice. This takes the form of a triple exponential function, the two fast decaying components representing the absorption of longer IR wavelengths at shallow depths, whilst the remaining one exhibits a decay constant of 2.04 m$^{-1}$, close to that previously noted in the literature. Our expression retains information

about the specificity of the ice optical properties by including the broadband albedo.

**Code availability**

Copies of the model routines are available from the corresponding author on request.



**Acknowledgements**

The authors would like to acknowledge the follow sources of support that facilitated this study and preparation of the subsequent paper: EPSRC MAPLE Platform Grant No. EP/I01912X/1 (GWE), a summer bursary from the Paneth Meteorite Trust (EH) and Grant No. RPG-2016-349 awarded by The Leverhulme Trust (ARDS, GWE).

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

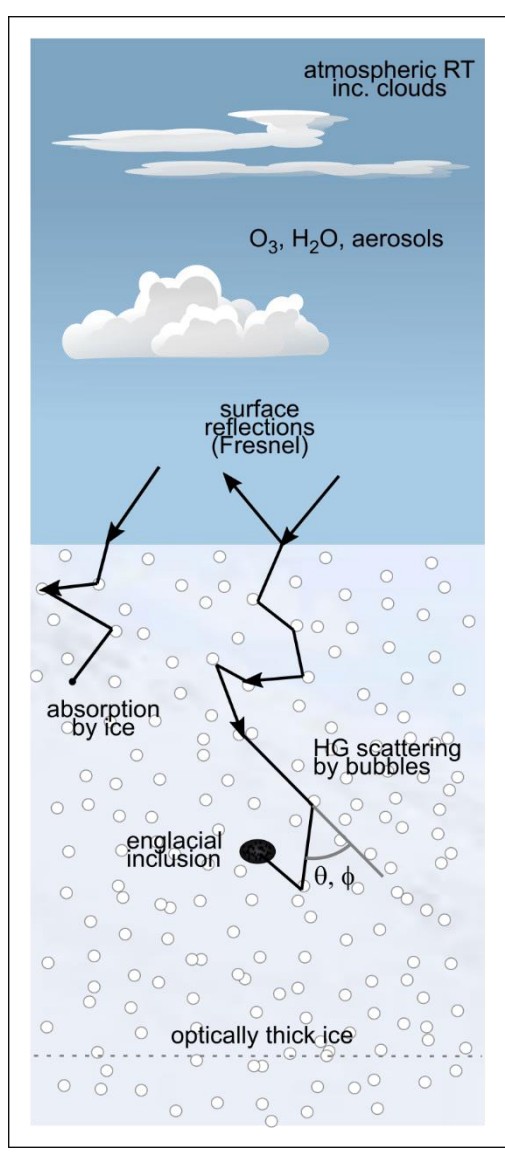

10    **Figure 1. Schematic illustrating model geometry (not to scale).**



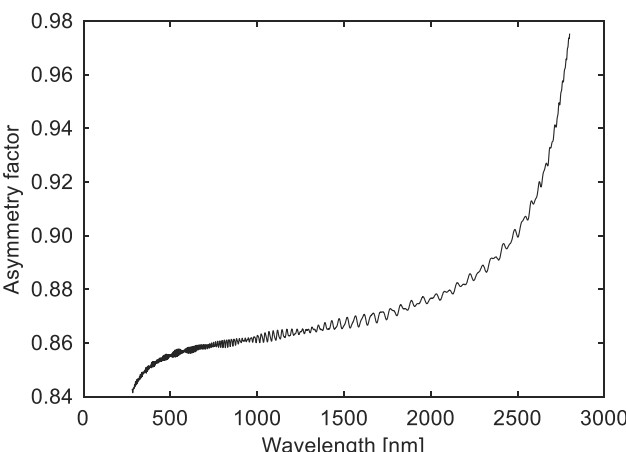

**Figure 2. Spectral variation of asymmetry parameter for spherical air bubbles in ice, calculated from Mie theory (Bohren and Huffman 1983) for a bubble radius of 198 μm.**





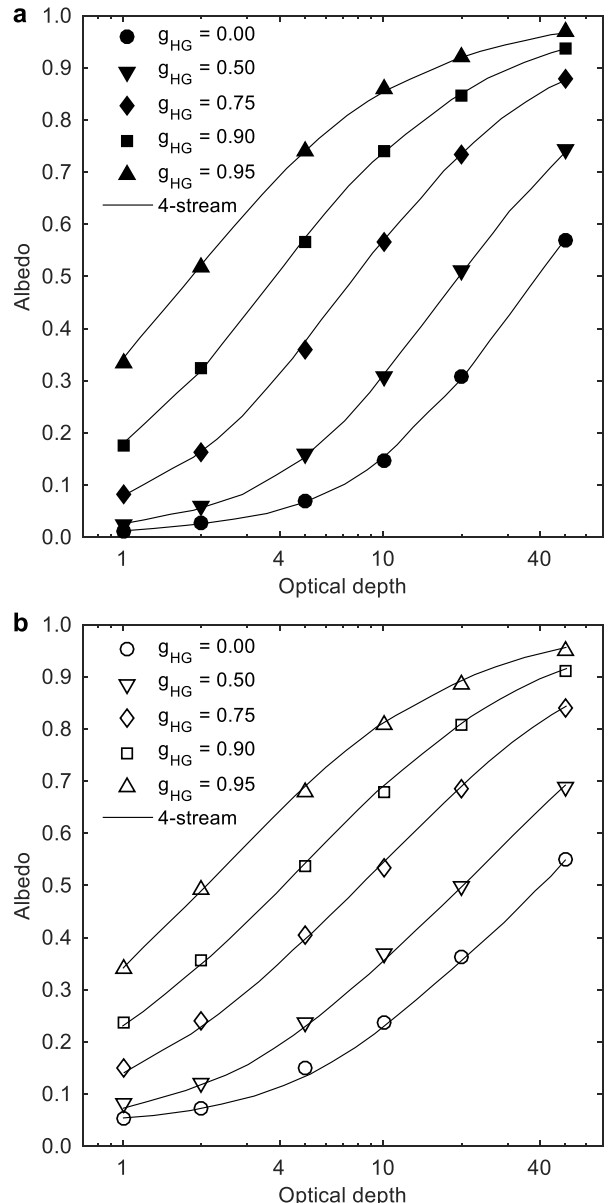

**Figure 3.** Comparison of albedo for (a) conservative non-refractive slab and (b) conservative refractive slabs as a function of asymmetry parameter and optical depths. Lines are digitised four-stream results taken from Light et al. (2003) and symbols indicate results calculated from the Monte Carlo code described here. All inputs are chosen to match those in Light et al. (2003).



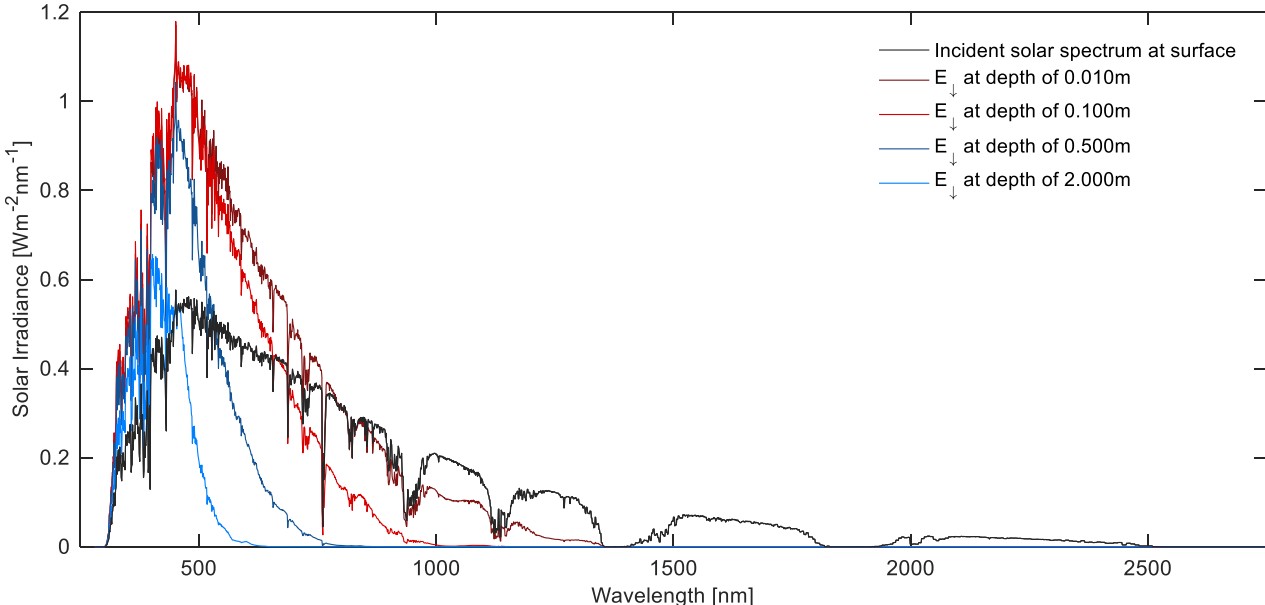

**Figure 4. Spectral downwelling irradiance at the ice surface and at selected depths. The incident solar spectrum is calculated as described in Sect. 2; within the ice, the effective bubble radius is 198 μm and the number concentration is 415 cm⁻³, corresponding to the Dadic BIAs unadjusted no cracks parameter set in Table 1.**

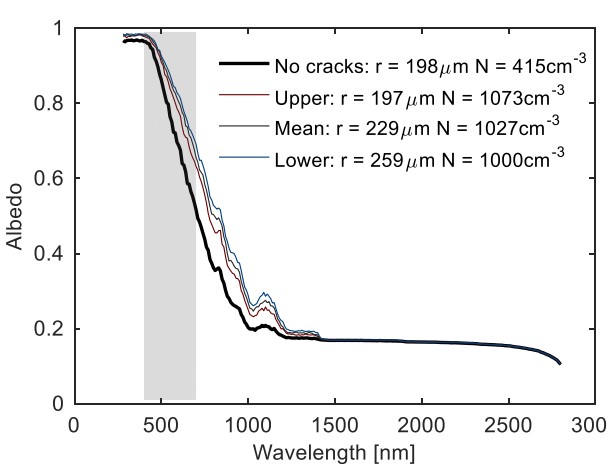

**Figure 5. Spectral albedo calculated for selected bubble parameter sets from Table 1, corresponding to porosities (bubble volume fractions) of 1.35 % for the original Dadic BIAs unadjusted no cracks parameter set ($r_{bub}$ = 198 μm, $N_{bub}$ = 415 cm⁻³); 5.27 % for the mean Dadic BIAs combined mCT and caliper parameter set ($r_{bub}$ = 229 μm, $N_{bub}$ = 1027 cm⁻³); 3.45 % for the upper Dadic**

10 **BIAs mCT density parameter set ($r_{bub}$ = 197 μm, $N_{bub}$ = 1073 cm⁻³); and 7.31 % for the lower parameter set corresponding to Bintanja's (1999) 850 kg m⁻³ density end point ($r_{bub}$ = 259 μm, $N_{bub}$ = 1000 cm⁻³). Grey shading indicates visible wavelengths.**





**Figure 6. (a) Fractional solar irradiance variation against ice depth for bubble parameter sets from Table 1. Solid lines show the downwelling irradiance, $E_\downarrow/E_0$, whilst dotted lines show the upwelling irradiance, $E_\uparrow/E_0$. The pairs of lines correspond to porosities (bubble volume fractions) of 1.35 % for the Dadic BIAs unadjusted no cracks parameter set ($r_{bub} = 198$ μm, $N_{bub} = 415$ cm$^{-3}$); 3.45 % for the upper Dadic BIAs mCT density parameter set ($r_{bub} = 197$ μm, $N_{bub} = 1073$ cm$^{-3}$); 5.27 % for the mean Dadic BIAs combined mCT and caliper parameter set ($r_{bub} = 229$ μm, $N_{bub} = 1027$ cm$^{-3}$); and 7.31 % for the lower parameter set corresponding to Bintanja's (1999) 850 kg m$^{-3}$ density end point ($r_{bub} = 259$ μm, $N_{bub} = 1000$ cm$^{-3}$). (b) Shows upper grey highlighted area to emphasize subsurface region of Fig. 6a. (c) Shows fractional net irradiance, $(E_\downarrow - E_\uparrow)/E_0$, for same bubble parameter sets, whilst (d) shows fractional mean irradiance, $(E_\downarrow + E_\uparrow)/2E_0$, in the area highlighted in Fig. 6a. In all cases $E_0 = 302.5$ W m$^{-2}$.**

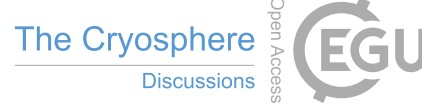



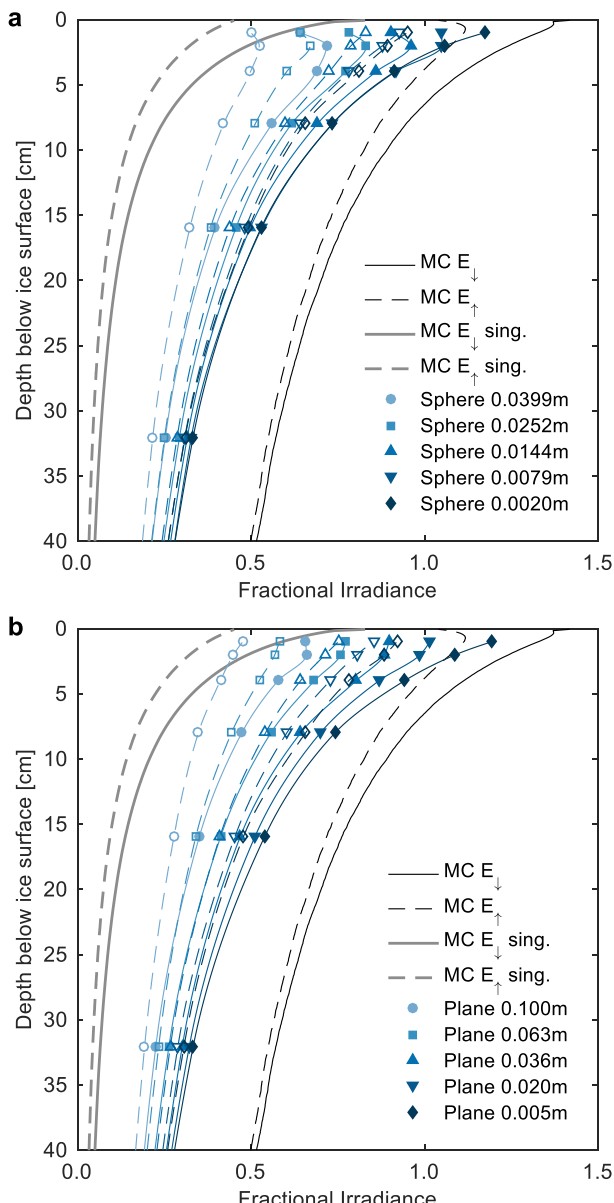

**Figure 7. (a) Fractional solar irradiance absorbed by spherical inclusions against ice depth for a fully diffuse sky and assuming the mean Dadic BIAs combined mCT and caliper bubble parameter set. The solid dark grey line indicates the fractional multiple scattered downwelling components (MC $E_\downarrow$, $E_\uparrow$) within the ice as in Fig. 6, the dotted grey line indicate the upwelling component. Their light grey counterparts show the singly scattered contributions (MC $E_\downarrow$, $E_\uparrow$ sing.). The area-adjusted upwelling and downwelling flux impacting on the ellipsoidal inclusions are shown as coloured markers, whilst the solid and dotted lines are interpolations between these. (b) Fractional solar irradiance absorbed by planar inclusions against ice depth for a fully diffuse sky and assuming the mean Dadic BIAs combined mCT and caliper bubble parameter set. Markers and lines are as in Fig. 7a.**



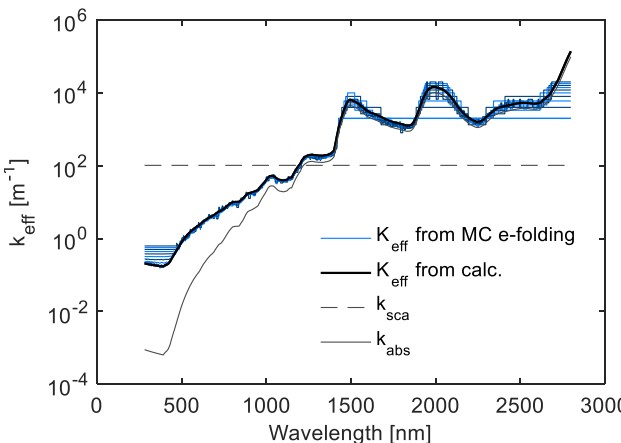

**Figure 8. Spectral variation of the effective attenuation coefficient calculated from e-folding distances (as described in the text) against that calculated from Marchesini et al. (1989).** $K_{eff}$ **from calculations is shown using a multiplier of 2 for the** $k_{abs}^2$ **term in Eqn 8. The e-folding distances are calculated as** $1/n$ **of the depth by which the Monte Carlo downwelling irradiance falls to** $e^{-n}$ **of its initial values where** $n$ **are integers in the range [1 … 16]. Also shown is the absorption coefficient for pure ice (** $k_{abs}$ **) and the scattering coefficient (** $k_{sca}$ **) for the Dadic BIAs unadjusted no cracks parameter set (** $V_f$ **= 1.35 %;** $r_{bub}$ **= 198 μm,** $N_{bub}$ **= 415 cm$^{-3}$).**

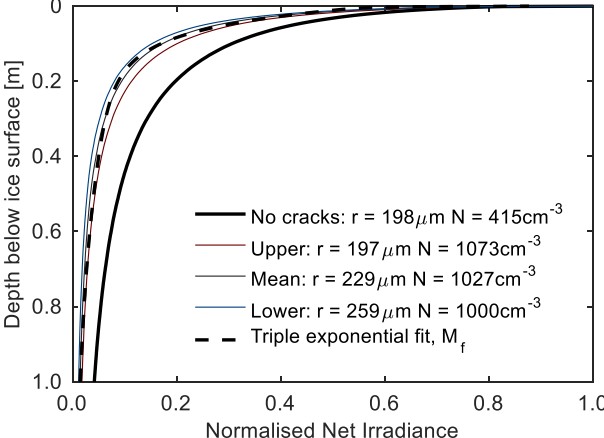

**Figure 9. This figure shows the scaled net irradiance for the four different bubble data sets. The dashed black line, which is almost identical to the grey, shows the triple exponential curve fit,** $M_f$**.**

**Table 1. Summary of bulk ice bubble data referred to in the text, ranked by increasing bulk density. The Bintanja (1999) (850 kg m$^{-3}$) parameter set is derived from Bintanja's lower estimate of blue ice density and is used as a lower bound parameter set in the text. The mean parameter set is the arithmetic mean of the bubble datasets described in Appendix B and listed in Table B1. The Dadic BIAs mCT density parameters select only samples with microCT density measurements, which are selected to avoid regions with cracks and is used as a parameter set corresponding to a realistic upper bound for density. The Dadic BIAs unadjusted parameter set is also calculated from samples without cracks, but relying on unadjusted microCT density measurements]. This last parameter set acts as an outer upper bound with especially low porosities and a density approaching that of pure ice; hereinafter it is referred to the no cracks parameter set.**



|  | Density [kg m$^{-3}$] | Porosity, $V_f$ [%] | $r_{bub}$ [mm] | $N_{bub}$ [mm$^{-3}$] | $k_{sca}$ [m$^{-1}$] |
|---|---|---|---|---|---|
| Bintanja (1999) (850 kg m$^{-3}$) [lower] | 850.0 | 7.31 | 0.259 | 1.000 | 422.7 |
| Mean | 868.7 | 5.27 | 0.229 | 1.027 | 340.5 |
| Dadic BIAs mCT density [upper] | 885.4 | 3.45 | 0.197 | 1.073 | 262.4 |
| Dadic BIAs unadjusted [no cracks] | 904.6 | 1.35 | 0.198 | 0.415 | 102.2 |





**Appendices**

**A Atmospheric Radiative Transfer**

In order to study the passage of solar radiation through bubbled ice, it is necessary to calculate the solar spectrum incident on that surface, considering that many aspects of the transfer are wavelength dependent. Here, the incident spectral irradiance at

the ice surface is calculated using the libRadtran radiative transfer model (Mayer and Kylling, 2005) with relevant atmospheric inputs. These inputs are broadly similar to those in Evatt et al. (2016) used to calculate a climatology of integrated SW fluxes, but for completeness we describe them in some more detail here. Internally we use the sdisort radiative transfer solver with pseudo-spherical approximation and the reptran molecular absorption parameterisation, to calculate surface spectral irradiance between 250 nm and 2800 nm at 1 nm intervals. The extra-terrestrial solar spectrum is from Kurucz (1994). The output altitude

is set at 2.04 km with a nominal albedo of 0.62, suitable for the BIA located near Frontier Mountain range in Antarctica [72.95º S, 160.48º W]. The solar zenith angle is set to 66.4º, a representative value that results in a spectrally-integrated irradiance equal to the daily mean on the day of the austral summer solstice.

The clear sky atmosphere profile used is the subarctic summer profile (Anderson et al., 1986) with a climatological total ozone column of 300 DU (Diaz et al., 2004). The spring-summer aerosol profile is taken from Shettle (1989) with the aerosol optical

depth (AOD) at 550 nm scaled to 0.028, the mean of the high and low elevation AOD estimates given by Tomasi et al. (2007). In order to include the effect of clouds, the model is run three times: once with no clouds with the parameters above, once with the addition of a high altitude glaciated cloud, and once with a lower level cloud. The three spectra are then combined linearly in proportion to the relative occurrence estimates of clear skies, high level and lower level clouds to form a single irradiance spectrum. Cloud heights, depths, occurrence frequencies and bulk microphysical properties are taken from Adhikari et al.

20    (2012).

Care has been taken to be select representative parameter values for the specific BIA locality. However it is anticipated that the resultant spectral shape, though not necessarily the absolute total irradiance, should be also generally applicable to high altitude polar regions.

**B Blue ice bubble data**

Once the incoming solar radiation reaches the ice surface, the key moderators of radiation through blue ice are the number concentration and radii of bubbles. As noted in the main text there is a paucity of relevant in-situ data and thus we principally rely on a homogenisation of bubble number concentration and radii measurements carried out by Dadic et al. (2013) near the Transantarctic Mountains, Antarctica. Dadic et al.'s samples cover a range of ice conditions along two BIA transects with sample depths down to 0.94 m and are analysed by a combination of microCT analysis, specific surface area (SSA) derived

estimates and caliper-and-scale measurements. As the authors state, the bulk ice densities from the caliper measurements are expected to be under-estimates of the density, whilst those from microCT analysis are expected to be over-estimates. To





reconcile these differences and formulate a best estimate of bubble radius and number concentration we first calculate the mean ratio of microCT and caliper densities for samples where both methods have been employed, which allows us to calculate an adjusted density estimate for all samples. In a similar fashion a simple regression is found between measured bubble radii and bulk ice density for the subset of samples undergoing both analysis methods, to give an estimate for samples lacking radii

measurements. In this way we formulate a self-consistent set of densities and radii for the 27 samples available, which together have a mean density of 875±12 kg m$^{-3}$ and a mean bubble radius of 0.236±0.028 mm. A bulk density of pure ice, $\rho_{pure}$, of 917 kg m$^{-3}$ is assumed, and the bulk density, $\rho_{ice}$, effective bubble radius, porosity (or volume fraction of bubbles) $V_{bub}$, and bubble number concentration are related by the following two expressions:

$$\rho_{ice} = \rho_{pure}(1 - V_{bub}) \tag{B1}$$

$$V_{bub} = \frac{4\pi r_{bub}^3 N_{bub}}{3} \tag{B2}$$

To ensure we are not overly reliant on a single set of field campaign data, we apply the density-radii regression and the usual density-porosity-number concentration relations to sub-samples of Dadic et al.'s data and estimates from the wider literature. In this way we aim to represent the range of environments present in BIAs, and are able to calculate an overall mean parameter set and choose example parameter sets corresponding to low and high bulk densities. We additionally select values

corresponding to an outer upper bound for blue ice density and referred to as the no cracks parameter set. Each of these four parameter sets are self-consistent between their estimates of effective bubble radius and number concentration, and, in line with observed bulk densities and porosity values. The parameter sets contributing to the overall mean are listed in Table B1, with additional data being noted in Table B2.

### C Inclusion (Meteorite) Dimensions

We extract dimensions from 94 recently discovered samples detailed in ANSMET newsletters (ANSMET, 2017); all meteorite data are combined to give a mean length × width × depth, which we interpret as the lengths of the principal axes $[a\ b\ c]$ of a tri-axial ellipsoid. We find a median value of $a$ to be 2.75 cm with interquartile range of 1.5 cm to 4.8 cm. We also calculate the ratios $b/a$ (0.775±0.015) and $c/a$ (0.502 ± 0.17) from which we define a representative lower quartile, a median, and, an upper quartile ellipsoid.

This defines three macroscopic ellipsoidal inclusions with upward facing surface areas of 4.00 cm$^2$, 13.4 cm$^2$, and 40.9 cm$^2$. To ensure the general conclusions are applicable to any convex absorbing inclusion within the ice, we add one larger target with the same aspect ratios, but an upper surface area of 100 cm$^2$, and one smaller with an upper surface area of 0.25 cm$^2$. Based on these ellipsoids, five planar and five spherical targets are also defined with linear dimensions and radii chosen so that their upward facing surface areas match those of the set of ellipsoids.




**Table B1.** Summary of bulk ice bubble data contributing to mean parameter set. Dadic BIAs caliper density parameters selects only samples directly measured by calipers (including samples designated as having cracks). Dadic BIAs combined mCT and caliper parameters incorporate a full set of samples, data being homogenised as described in the text of Appendix B. Dadic BIAs mCT density parameters selects only samples with microCT density measurements (chosen to avoid regions with cracks) selected as representing realistic upper bound for density. Mellor and Swithinbank (1989) parameters are constructed from an estimate that BIAs have a porosity of 6%, combined with the bubble radii-density regression noted in the accompanying text. The Bintanja 1999 (850 kg m$^{-3}$) values are derived from Bintanja's lower estimate of blue ice density and is selected as lower parameter set. The Bintanja (1999) (865 kg m$^{-3}$) parameter set is derived from the mid-point of Bintanja's range estimate of blue ice density whilst the Bintanja (1999) (880 kg m$^{-3}$) parameter is derived from upper limit for blue ice density. The mean parameter set is the arithmetic mean of these preceding estimates.

| | Density [kg m$^{-3}$] | Porosity, $V_f$ [%] | $r_{bub}$ [mm] | $N_{bub}$ [mm$^{-3}$] | $k_{sca}$ [m$^{-1}$] |
|---|---|---|---|---|---|
| Dadic BIAs caliper density | 863.3 | 5.86 | 0.236 | 1.064 | 372.3 |
| Dadic BIAs combined mCT and caliper | 875.1 | 4.57 | 0.236 | 0.834 | 290.7 |
| Dadic BIAs mCT density [upper] | 885.4 | 3.45 | 0.197 | 1.073 | 262.4 |
| Mellor and Swithinbank (1989) | 862.0 | 6.00 | 0.238 | 1.059 | 377.7 |
| Bintanja (1999) (850 kg m$^{-3}$) [lower] | 850.0 | 7.31 | 0.259 | 1.000 | 422.7 |
| Bintanja (1999) (865 kg m$^{-3}$) | 865.0 | 5.67 | 0.233 | 1.070 | 365.1 |
| Bintanja (1999) (880 kg m$^{-3}$) | 880.0 | 4.04 | 0.207 | 1.091 | 292.8 |
| Mean | 868.7 | 5.27 | 0.229 | 1.027 | 340.5 |

**Table B2.** Summary of additional bulk ice data bubble data not contributing to mean parameter set in Table B1. Dadic BIAs unadjusted [no cracks] parameter set calculated from samples classed as not including cracks, and relying on unadjusted microCT density measurements. Dadic BIAs mCT density (SSA adjusted) is derived from Dadic BIAs mCT in Table B1, but adjusted for SSA attributed to cracks vs no-crack proportions. Dadic BIAs crack regions selects crack only regions, with a consequently low bulk density.

| | Density [kg m$^{-3}$] | Porosity, $V_f$ [%] | $r_{bub}$ [mm] | $N_{bub}$ [mm$^{-3}$] | $k_{sca}$ [m$^{-1}$] |
|---|---|---|---|---|---|
| Dadic BIAs unadjusted [no cracks] | 904.6 | 1.35 | 0.198 | 0.415 | 102.2 |
| Dadic BIAs mCT density (SSA adjusted) | 866.8 | 5.48 | 0.197 | 1.703 | 416.5 |
| Dadic BIAs crack regions | 732.0 | 20.17 | 0.237 | 3.618 | 1276.7 |