# Peer review of "Solar SW radiative transfer in bubbled ice: spectral considerations, subsurface enhancement, and inclusions"

_The Cryosphere, 2018_

## Referee Comment (RC1) · S. Warren (Referee) · 19 Jun 2018

Because of the refractive interface at the surface of blue-ice fields in Antarctica, the authors expect a "subsurface enhancement in both the downwelling and upwelling fluxes relative to the incidence irradiance." This subsurface enhancement is a consequence of total internal reflection for angles of upward radiation greater than about 50 degrees. The authors have exaggerated this enhancement in two ways.

(1) The irradiance is the integral of radiance over the $2\pi$ solid angle of a hemisphere, with weighting by $\cos\theta$, where $\theta$ is the angle of incidence. As a consequence, the reflectance $\bar{R}$ for diffuse incidence is $\bar{R} = \dfrac{\int R(\theta)\cos\theta\, d\omega}{\int \cos\theta\, d\omega} = \dfrac{\int R(\theta)\cos\theta\sin\theta\, d\theta}{\int \cos\theta\sin\theta\, d\theta}$. The weighing factor $\cos\theta$ appears to be missing from the authors' calculation. On page 8 line 30, for a refractive index m=1.306 and diffuse incidence, the authors get a transmittance T=0.832, implying a reflectance R = 1-T = 0.17. The correct value of reflectance for diffuse incidence, computed using the above equation, is R=0.06 (which by the way is confirmed experimentally as the albedo of a flat water surface under diffuse incidence), but the erroneous value R=0.17 could be obtained if the cosine-weighting was missing from the calculation. Also, on line 31, for the radiation incident on the ice/air interface from below (refractive index m = 1/1.306 = 0.776), the authors get R=0.668. The correct value is R=0.48, but the erroneous value R=0.668 could be obtained if the cosine-weighting was missing. This error must be corrected before resubmission.

(2) Secondly, the authors have exaggerated the enhancement by assuming the ice surface is planar. Glacier-ice surfaces of the Antarctic blue-ice fields are rough; it's a bumpy ride in a snowmobile. As a result, few photons coming from below will be incident on surface facets at local angles experiencing total internal reflection. Since the authors are using a Monte Carlo model, they can incorporate surface roughness. Perhaps there are some measurements of surface roughness that could guide the authors. If such measurements are not available, I suggest assuming a distribution of slope angles with a standard deviation of 20 degrees.

Minor comments:
Page 2 lines 12-13. "there is a tendency to treat the shortwave radiative flux as a single broadband parameter". For snow, the errors caused by the broadband approximation were shown by Brandt and Warren 1993 (*J. Glaciol., 39*, 99-110).

Page 4 line 14. The location given for Frontier Mountain (73 S, 160 W) is far out in the ocean. Probably the authors instead mean 160 E.

Page 4 line 22. Equation (1) is wrong. The denominator, $N_{bub}$, should instead be in the numerator, to give the correct units ($m^{-1}$) for $k_{sca}$.

Page 4 line 23. Equation (2) should be scaled by (1 − porosity).

---

## Referee Comment (RC2) · Anonymous Referee #2 · 20 Jul 2018

The manuscript is dedicated to the ray-tracing model of Antarctic blue ice with bubble inclusions. Ray-tracing models allow to obtain exact solutions of light scattering within particulate media with respect to complicated particle and inclusion shapes and size distributions, surface properties, various illumination conditions, etc and remain a valuable tool in understanding of light scattering processes within atmosphere and surface of the Earth and other planets.

In the presented manuscript, a non-polarized spectrally dependent ray tracing model of the plane parallel Antarctic blue ice with spherical air bubbles and inclusions is developed. While it certainly is an idealisation due to the mentioned assumptions, the

manuscript provides an insight into the energy distribution in the ice within quite some range of wavelengths, which is potentially important for a number of land ice and sea ice related applications. The used assumptions e.g. about the planar upper surface of the ice or the uniform distribution of spherical bubbles within the ice, although unrealistic, do help highlight the mechanisms responsible for specific light scattering scenarios which would not be observable otherwise. The presented ray tracing code should be easy to modify to account for e.g. surface roughness and the scientific community would certainly gain if the authors decide to continue and do so as a next step; the reviewer, however, appreciates the fundamental study on the subsurface enhancement because even if not always observable in the field due to many factors, the ray scenarios responsible for this enhancement may still be important and it is crucial to understand those scenarios.

The manuscript is well-written and well-structured, nice quality of the figures.

This work is certainly relevant, fits into the scope of the journal, and is recommended for the publication in The Cryosphere subject to minor corrections which are listed below.

- although the manuscript is quite comprehensive already, the reviewer would still like to suggest a short analysis on the orders of scattering and light ray scenarios which were responsible for the subsurface enhancement and peak. The corresponding Section 3.1 currently omits this and could be extended. Conversely, the self-shadowing explanation in the Section 4, Line 7-17, could be more condensed, so that the total volume of the manuscript does not need to increase.

- please take care to specify the important details on the model already in the abstract, e.g. the size parameter range, the fact that the model does not account for polarisation, the Henyey-Greenstein approximation, spherical bubbles, and plane surface.

- please give a short overview (from the perspective of formation and optical properties) on what the Antarctic blue ice is, e.g. in the Introduction, from the top of Page 2, or near Line 31 on P2, and how it (optically) differs from snow, sea ice, lake ice, Antarctic

firn with respect to layering and granulation.

- P6, Lines 27 - 30. Please consider making the description more self-consistent, so that referring to Light et al would be needed for additional information only.

- P7 Line 2 and Caption of Fig. 3. Albedo at which wavelength? please specify.

- Section 2.5 - here again please specify how the assumption of a continuous medium with spherical bubbles really relates to the Antarctic blue ice and other cryospheric surface types.

- P 7, Line 21-23: please reformulate the sentence or split into two, unclear what is meant here.

- please consider reformulating the title of the paper so that Monte-Carlo and Antarctic blue ice are included. They are the main scope of the manuscript and clarity is essential here, even when the results can potentially be applied in a broader context.

---

## Author Comment (AC1) · 20 Sep 2018

We thank both reviewers for their useful comments, which are addressed in turn below. Reviewers' comments are in italics, our responses in regular text and places where the manuscript has been altered are noted.

**RC1: Reviewer #1**

*Because of the refractive interface at the surface of blue-ice fields in Antarctica, the authors expect a "subsurface enhancement in both the downwelling and upwelling fluxes relative to the incidence irradiance." This subsurface enhancement is a consequence of total internal reflection for angles of upward radiation greater than about 50 degrees. The authors have exaggerated this enhancement in two ways.*

*(1) The irradiance is the integral of radiance over the $2\pi$ solid angle of a hemisphere, with weighting by $\cos\theta$, where $\theta$ is the angle of incidence. As a consequence, the reflectance $\bar{R}$ for diffuse incidence is*

$$\bar{R} = \frac{\int R(\theta)\cos\theta \, d\omega}{\int \cos\theta \, d\omega} = \frac{\int R(\theta)\cos\theta\sin\theta \, d\theta}{\int \cos\theta\sin\theta \, d\theta}$$

*The weighing factor $\cos\theta$ appears to be missing from the authors' calculation. On page 8 line 30, for a refractive index $m = 1.306$ and diffuse incidence, the authors get a transmittance $T = 0.832$, implying a reflectance $R = 1 - T = 0.17$. The correct value of reflectance for diffuse incidence, computed using the above equation, is $R = 0.06$ (which by the way is confirmed experimentally as the albedo of a flat water surface under diffuse incidence), but the erroneous value $R = 0.17$ could be obtained if the cosine-weighting was missing from the calculation. Also, on line 31, for the radiation incident on the ice/air interface from below (refractive index $m = 1/1.306 = 0.776$), the authors get $R = 0.668$. The correct value is $R = 0.48$, but the erroneous value $R = 0.668$ could be obtained if the cosine-weighting was missing. This error must be corrected before resubmission.*

Both the reflection ($R$) and transmission ($T$) calculation on p8 that reviewer 1 refers to, and the core model from which it derived, are based on a Monte Carlo model, tracing photon paths. In both cases we do not weight the result by $\cos\theta$ when calculating irradiance or related quantities. With a measurement background we – like the reviewer – initially expected to have to include a cosine weighting, but this is a misconception.

If we consider a photon (packet), with an energy $dQ$ passing through a unit surface area $dA$ at an angle $\theta$ to the surface normal within a solid angle $d\Omega$ in time $dt$, then the associated radiance $L$ (energy per unit time, per unit area, per unit solid angle) is (Wood et al 2001, Woan 2000):

$$L = \frac{dQ}{\cos\theta \, dA \, dt \, d\Omega}$$

Further, irradiance is defined as:

$E = \int L\cos\theta \, d\Omega$,

with the contribution of an individual photon to the irradiance then given as:

$dE = L\cos\theta \, d\Omega$.

Combining the first and last expressions, we have:

$dE = \frac{dQ}{dA \, dt}$.

Hence, when carrying out Monte Carlo calculations, the irradiance ($E$), and by extension the reflectance, are found from summing the number of photons each of energy $dQ$ passing through a surface area $dA$ in time $dt$, with no cosine weighting applied. We have clarified this at p7 l1-2.

The lack of need for a cosine weighting when calculating irradiance was verified prior to this study by comparison of a related atmospheric Monte Carlo code against the well-documented, and experimentally validated, libRadtran model. It was also confirmed by one of libRadtran's developers (B. Mayer, pers. comm., 2007).

It is perhaps worth noting that if a cosine weighting were needed, this would lead to an increase in the subsurface peak as it is in this region that the $\overline{\cos\theta}$ takes its largest value. Were a cosine weighting applied for the reflectance calculations but not the core Monte Code, this would lead to inconsistencies between the theoretical limit and modelled results as discussed at the end of Sect. 3.1.

As to the discrepancy between large scale measurements of water's albedo (6 to 10%) and our calculated reflectance value of 17%, there are a number of contributing factors. First our result is idealised, calculated for a flat surface with a perfectly diffuse incident radiation field. It agrees with the result of Schmidt (1915) noted by Neiburger (1942). Measurements of water reflectance are typically lower for a range of reasons (Neiburger, 1942; Katsaros et al, 1985). First the incident radiation field is seldom fully diffuse and with a direct component and solar elevation > 30°, the reflectance will be lower (we note that in Katsaros et al.'s Fig.3 as the total atmospheric transmission decreases, the band A albedo approaches a value of approximately 16-17%). Second, due to surface waves any field measurement of a water surface is likely to exclude reflections at higher incident angles which disproportionately contribute to larger reflectance values. Finally measurements are ultimately reliant on the quality of the detector optics response (and cosine optics are notoriously poor at glancing angles close to 90°). Any of these considerations would act to reduce the observed value. Indeed if we rerun our calculations of $R$ but reject photons at glancing angles of 12° or less, this alone is sufficient to produce a reflectance of 6% rather than 17% – and hence our model results are consistent with previous observations.

We acknowledge reviewer 1's point that natural surfaces are not perfectly flat, and therefore in future studies we plan to include this aspect into our model.

Katsaros, K. B., McMurdie, L. A., Lind, R. J. and Devault, J. E.: Albedo of a water surface, spectral variation, effects of atmospheric transmittance, sun angle and wind speed., *J. Geophys. Res.*, **90**(C4), 7313–7321, doi:10.1029/JC090iC04p07313, 1985.
Neiburger, M.: The reflection of diffuse radiation by the sea surface, *Am. Geophys. Union Trans.*, **29**(5), 647–652, 1942.
Schmidt, W.: No Strahlung und Verdunstung an frelan Wasserflachen, *Ann. Hydro.*, **43**, 169–178, 1915.
Woan, G.: *The Cambridge Handbook of Physics Formulas*, Cambridge University Press, 2000.
Wood, K., Whitney, B., Bjorkman, J. and Wolff, M.: *Introduction to Monte Carlo radiation transfer*, 2001.

—

*(2) Secondly, the authors have exaggerated the enhancement by assuming the ice surface is planar. Glacier-ice surfaces of the Antarctic blue-ice fields are rough; it's a bumpy ride in a snowmobile. As a result, few photons coming from below will be incident on surface facets at local angles experiencing total internal reflection. Since the authors are using a Monte Carlo model, they can incorporate surface roughness. Perhaps there are some measurements of surface roughness that could guide the authors. If such measurements are not available, I suggest assuming a distribution of slope angles with a standard deviation of 20 degrees.*

We accept that this is a limitation of the current idealised study, and it is accordingly noted in Sect. 2.5. However we also agree that slope aspect and roughness are certainly aspects worthy of investigation as we extend the model in the future.

—

*Minor comments:*

*Page 2 lines 12-13. "there is a tendency to treat the shortwave radiative flux as a single broadband parameter". For snow, the errors caused by the broadband approximation were shown by Brandt and Warren 1993 (J. Glaciol., 39, 99-110).*

The suggested reference has been added to the text and reference list.

—

*Page 4 line 14. The location given for Frontier Mountain (73 S, 160 W) is far out in the ocean. Probably the authors instead mean 160 E.*

The longitude has been corrected at p4 l26, p8 l14, and p27 l11

—

*Page 4 line 22. Equation (1) is wrong. The denominator, $N_{bub}$, should instead be in the numerator, to give the correct units ($m^{-1}$) for $k_{sca}$.*

Equation (1) has been corrected.

—

*Page 4 line 23. Equation (2) should be scaled by (1 – porosity)*

This oversight has been corrected in the text at p5, ll4-5.

**RC2: Reviewer #2**

*The manuscript is dedicated to the ray-tracing model of Antarctic blue ice with bubble inclusions. Ray-tracing models allow to obtain exact solutions of light scattering within particulate media with respect to complicated particle and inclusion shapes and size distributions, surface properties, various illumination conditions, etc and remain a valuable tool in understanding of light scattering processes within atmosphere and surface of the Earth and other planets.*

*In the presented manuscript, a non-polarized spectrally dependent ray tracing model of the plane parallel Antarctic blue ice with spherical air bubbles and inclusions is developed. While it certainly is an idealisation due to the mentioned assumptions, the manuscript provides an insight into the energy distribution in the ice within quite some range of wavelengths, which is potentially important for a number of land ice and sea ice related applications. The used assumptions e.g. about the planar upper surface of the ice or the uniform distribution of spherical bubbles within the ice, although unrealistic, do help highlight the mechanisms responsible for specific light scattering scenarios which would not be observable otherwise. The presented ray tracing code should be easy to modify to account for e.g. surface roughness and the scientific community would certainly gain if the authors decide to continue and do so as a next step; the reviewer, however, appreciates the fundamental study on the subsurface enhancement because even if not always observable in the field due to many factors, the ray scenarios responsible for this enhancement may still be important and it is crucial to understand those scenarios.*

*The manuscript is well-written and well-structured, nice quality of the figures.*

*This work is certainly relevant, fits into the scope of the journal, and is recommended for the publication in The Cryosphere subject to minor corrections which are listed below.*

*Although the manuscript is quite comprehensive already, the reviewer would still like to suggest a short analysis on the orders of scattering and light ray scenarios which were responsible for the subsurface enhancement and peak. The corresponding Section 3.1 currently omits this and could be extended. Conversely, the self-shadowing explanation in the Section 4, Line 7-17, could be more condensed, so that the total volume of the manuscript does not need to increase.*

There is a discussion of the scattering process that leads to the subsurface enhancement given at p8 ll29 to p9 l8. We have added reference to orders of scattering within this and comments on the process leading to the peak itself are included in Sect. 3.2. As the manuscript length has not been increased much by this change, the description of self-shadowing has been left unaltered.

—

*Please take care to specify the important details on the model already in the abstract, e.g. the size parameter range, the fact that the model does not account for polarisation, the Henyey-Greenstein approximation, spherical bubbles, and plane surface.*

These additional details of the model have been included in the abstract at p1, ll9-10, or else added at the relevant point in the text (p4 l3, p6 ll17-19).

—

*Please give a short overview (from the perspective of formation and optical properties) on what the Antarctic blue ice is, e.g. in the Introduction, from the top of Page 2, or near Line 31 on P2, and how it (optically) differs from snow, sea ice, lake ice, Antarctic firn with respect to layering and granulation.*

Following the reviewers suggestion a paragraph on the optical differences between blue ice and other cryospheric forms has been added at p2, ll19-29.

—

*P6, Lines 27 - 30. Please consider making the description more self-consistent, so that referring to Light et al would be needed for additional information only.*

These sentences have been expanded to be more self-consistent and now include a statement of the parameters used by Light et al. Edits have been performed at p7 ll10-17.

—

*P7 Line 2 and Caption of Fig. 3. Albedo at which wavelength? please specify.*

Light et al's validation tests did not specify a wavelength per se, but instead set the real part of refractive index to 1.31. This corresponds to a nominal wavelength of 580 nm in the visible range.

—

*Section 2.5 - here again please specify how the assumption of a continuous medium with spherical bubbles really relates to the Antarctic blue ice and other cryospheric surface types.*

These assumptions have been detailed and expanded upon between p7 l27 and p8 l3.

—

*P7, Line 21-23: please reformulate the sentence or split into two, unclear what is meant here.*

The sentence has been split and clarified at p8 ll7-10.

*Please consider reformulating the title of the paper so that Monte-Carlo and Antarctic blue ice are included. They are the main scope of the manuscript and clarity is essential here, even when the results can potentially be applied in a broader context.*

We thank the reviewer for the suggestion. The title has been reformulated as "Solar radiative transfer in Antarctic blue ice: spectral considerations, subsurface enhancement, and inclusions", and whilst for reasons of brevity Monte Carlo does not appear in the title, we note it is mentioned in the first sentence of the abstract.

In addition we have made some small typographic corrections throughout, including the reference list.